# Structure characteristics of mutation sites in two *waxy* alleles from Yunnan waxy maize (*Zea mays* L. var. *certaina* Kulesh) landraces

**Tingting Sun, Xiaoyang Wu** *

Biotechnology and Germplasm Resources Institute, Yunnan Academy of Agricultural Sciences, Kunming, China

* xtwxy1@163.com

## Abstract

A large number of waxy maize landraces are distributed in Yunnan and surrounding areas, and abundant *waxy* alleles of different types are distributed in these landraces. The identification of *waxy* alleles is helpful to the protection and utilization of these waxy landraces. This study introduced structure characteristics of *waxy* genes from two specific landraces of Yunnan, Zinuoyumi and Myanmar Four-Row Wax. Zinuoyumi has two *waxy* alleles *wx-Cin4* and *wx-Cin4-2*; Myanmar Four-Row Wax has three *waxy* alleles *wx-D10*, *wx-Reina* and *wx-D11*. The *wx-Cin4-2* and *wx-D11* are two types of *waxy* alleles first reported in this study. The *wx-Cin4-2* has two mutation sites, deletion of 30 bp in exon 10, insertion of a 1,267 bp non-long terminal repeat (non-LTR) retrotransposon *Cin4* in intron 10, and 13 bp extra sequence were found at 5' end of the *Cin4*; the mutation site of *wx-D11* is a 1,082 bp deletion from exons 11 to 14 of the *waxy* gene and is replaced with a 72 bp filler sequence. This study enriched the type of *waxy* allele from Yunnan waxy maize landraces and further discussed the molecular basis for the formation of mutation sites of *wx-Cin4-2* and *wx-D11*.

## Introduction

Maize (*Zea mays* L.) originated in South America and was introduced to China ~400–500 years ago [1]. In Southwest China, residents have the habit of waxy food. Maize is under selection pressure since the introduction, and then waxy maize (*Zea mays* L. var. *certaina* Kulesh) forms a branch of maize [2]. Amylopectin is almost all starch in the endosperm of waxy maize, which makes the grain have a waxy taste [3, 4]. The waxy taste is controlled by a recessive *waxy* gene. The maize *waxy* gene is located on chromosome 9, with a total length of 4.5-Kb and contains 14 exons [5]. In the starch synthesis pathway, the *waxy* gene expresses Granule-Bound Starch Synthase I (GBSS I) protein for amylose synthesis; and the *waxy* gene is a key gene affecting grain quality. Yunnan is the province with the most ethnic groups in China, which is located on the Yunnan-Guizhou Plateau, bordering Myanmar, Laos and Vietnam. There is a view that Yunnan and its surrounding areas are the origin center and genetic diversity center of Chinese waxy maize [6–9]. There are abundant waxy maize landraces [10], such as an

**Data Availability Statement:** The gene sequence in this article has been submitted to the genbank database and the sequence number wx-Cin4-2

(OR161363) wx-D11 (OR161364) has been obtained.

**Funding:** This work was supported by the National Natural Science Foundation of China (32060459). The funders had no role in study design, data collection and analysis, decision to publish, or preparation of the manuscript.

**Competing interests:** The authors have declared that no competing interests exist.

ancient landrace termed Four-row Wax, which has only four rows of seed set in the cob and many characters similar to those of wild species. It was collected from Menghai County of Yunnan in 1970 and has been planted by the local Dai minority since 1890 [7, 11].

Abundant *waxy* alleles are distributed in these landraces of Yunnan (Fig 1). The known alleles include *wx-D7*, *wx-D10*, *wx-Cin4*, *wx-124*, *wx-Reina*, *wx-Xuanwei*, *wx-PIF/Harbinger*, *wx-hAT*, and *wx-Elote2* [12–15]; *wx-D7* and *wx-D10* were mutations caused by sequence deletion in *waxy* gene [16, 17]; *wx-Cin4*, *wx-124*, *wx-Reina*, *wx-Xuanwei*, *wx-PIF/Harbinger*, *wx-hAT*, and *wx-Elote2* are mutations caused by transposon insertion in the *waxy* gene [12–15]; *wx-D10* and *wx-Reina* are widely distributed *waxy* alleles in waxy maize landraces of Yunnan area [15, 18]. The *wx-D10* mutation site is a 30 bp deletion in the tenth exon of the *waxy* gene [16, 17]. The *wx-Reina* mutation site is the insertion of a 5.4 Kb long terminal repeat (LTR) retrotransposon *Reina*, into the tenth intron of the *waxy* gene [13].

It is necessary to prevent the loss of rare alleles during the preservation and reproduction of germplasm resources. Identification of alleles is one of the effective measures to prevent the loss of rare alleles. So the identification of *waxy* alleles is helpful to the protection and utilization of these waxy landraces. So far, a large number of Yunnan waxy maize landraces still contain unknown *waxy* alleles that need to be identified. In this study, we identified and introduced structure characteristics of two rare *waxy* alleles from two specific landraces of Yunnan, Zinuoyumi and Myanmar Four-Row Wax.

## Materials and methods

### Plant material

In this study, 405 waxy maize landraces from different regions of Yunnan were used as research material mentioned in our previous work [15]. The maize inbred line B73 was used as the non-waxy maize control. Seeds of research material were sampled from the Yunnan Provincial Crops Genebank of Yunnan Academy of Agricultural Sciences, P.R.China, and planted at Songming experimental field. I$_2$/KI staining was used to identify the waxy grain of maize [19].

### Identification of mutation sites of *waxy* gene

Genomic DNA was extracted from young leaves and used for *waxy* gene detection [20]. Specific markers are used to identify known *waxy* alleles (Table 1); Molecular markers that can cover the entire *waxy* gene are used to identify unknown *waxy* alleles (Table 2) same as our previous work [13]. PCR products were used for 1% agarose gel electrophoresis and direct sequencing.

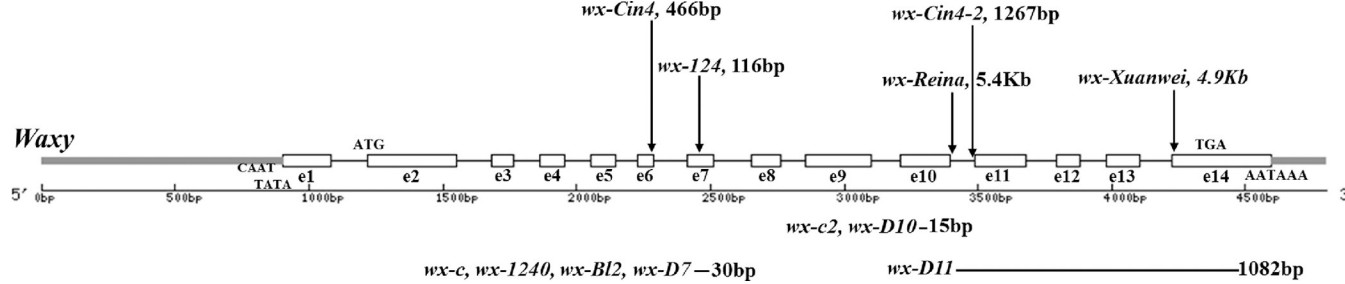

**Fig 1. Known *waxy* alleles in waxy maize landraces.** In maize, the wild-type *waxy* gene is about 4.5 Kb and has 14 exons numbered e1 to e14. The arrows above the schematic of the gene point to insertion mutations; the lines below indicate deletion mutations. Allele *wx-Cin4-2* is a 1,267 bp insertional mutation in intron 10. Allele *wx-D11* is a 1,082 bp deletion mutation from exon 11 to exon 14.

Table 1. Specific molecular markers for detecting different waxy alleles.

| Allele | Left primer sequence (5'-3') | Right primer sequence (5'-3') | Tm (°Cs) | Marker type | Reference |
|---|---|---|---|---|---|
| wx-D7 | TCGTGCTACCTCAAGAGCAA | TCTCAGAGAAGCACCACCAC | 57 | Co-dominant | [13] |
| wx-D10 | GTGGTGTGGTGTCCGGTT | TCTGCTCTTCCAGCCTGC | 57 | Co-dominant | [13] |
| wx-Cin4 | AGGATCCTGAGCCTCAACAA | TTCACCAATTGCGTAACCTG | 57 | Dominant | [15] |
| wx-124 | TACGAGACGGTCAGGTTC | GGTAGGAGATGTTGTGGAT | 55 | Co-dominant | [12] |
| wx-Reina | TCACCGTCAGCCCCTACTAC | CTACTGGGGTTTTGACAGTGG | 57 | Dominant | [13] |
| wx-Xuanwei | TTGGCCGACTCTACTGGTTT | TTCTCCCAGTTCTTGGCAGT | 57 | Dominant | [14] |
| wx-PIF/ Harbinger | GGTGACCAGAAGCTGCAAA | CGTGAAGCCTAGTGCCAAA | 57 | Dominant | [15] |
| wx-hAT | CCGAAGTTTTCGGTTTTCAT | CAAAGTATTCGGCCCACTGT | 57 | Dominant | [15] |
| wx-Elote2 | AGCAGTAGAGGCGCAAGAAG | GGCTCCCAACTACCATGAAA | 57 | Dominant | [15] |
| wx-Cin4-2 | GATCGTTCTGCTGGTACGTGT | AGTTCCTCACCATCTCCTCGT | 57 | Co-dominant | In this study |
| wx-D11 | TCACCGTCAGCCCCTACTAC | AACACCGAACAGCAGGGATT | 57 | Co-dominant | In this study |

## Sequence analysis

Gene structure was drawn using Gene Structure Display Server (GSDS) software http://gsds1.cbi.pku.edu.cn/ [21, 22]. BlastN in NCBI is used to analyze the type of insertion sequences and find reference sequences, and CDD is used to analyze the domain of transposons [23]. ORFfinder in NCBI is used to analyze the open reading frame (ORF). Soft-ware Clustal X (1.8) was used for sequence alignment [24]. The genome sequence of the maize inbred line, B73 (http://ftp.maizesequence.org/current/assembly/) [25], was used to identify copy number of transposon with blast tool (version 2.2.23+) [26]. The secondary structure of the *waxy* gene was predicted using RNA structure 6.2 software [27]; the neighbor-joining (NG) tree was constructed using Mega 5 software [28]. The primer design software was Primer3 [29].

## Results

This study identified the *waxy* allele composition of the Yunnan-specific waxy maize landraces Zinuoyumi and Four-Row Wax.

## Zinuoyumi

Zinuoyumi from Lancang County, Yunnan Province, except purple seed and cob, the purple pigment was deposited in the whole plant during the adult stage. It is an ideal purple gene source in black waxy maize breeding (Fig 2). It is the typical waxy maize with purple seed and cob among all landraces of purple waxy maize in Yunnan, so it is selected as one of the specific landraces used in this study.

Zinuoyumi has two *waxy* alleles, *wx-Cin4* and *wx-Cin4-2*. Unlike the known *waxy* alleles, *wx-Cin4-2* has two different mutation sites, 30 bp deletion in exon 10 same as *wx-D10* and 1,267 bp insertion of *Cin4* transposon in intron 10 (Fig 1). However, the allele *wx-D10* was not found in this material. It is the first time that *waxy* alleles with two mutation sites have been identified in waxy maize landraces of the Yunnan area.

*Cin4* is a class of non-long terminal repeat (non-LTR) retrotransposon in the maize genome [30]. The full-length *Cin4* transposon is about 7 Kb in length and has two open reading frames (ORF) to encode the necessary enzymes for jumping in the maize genome. Non-full-length *Cin4* is usually truncated at the 5'-end of the full-length *Cin4*. Both full-length and non-full-length *Cin4* have target site duplications (TSDs) sequence and poly (A) tail at 3'-end. The length of *Cin4* in *wx-Cin4-2* is 1,267 bp, which is a truncated transposon of *Cin4* (Fig 3); The

**Table 2. Molecular markers that can cover the entire waxy gene used in this study.**

| Primer name | Left primer sequence (5'-3') | Right primer sequence (5'-3') | Tm (˚C) |
|---|---|---|---|
| *waxy_1* | GGAACGGACTACAGGGACAA | ATGAGCTCCTCGGCGTAGTA | 57 |
| *waxy_3* | TCACCGTCAGCCCCTACTAC | CTTGCCTGGGAACTTCTCCT | 57 |
| *waxy_11* | GATCGTTCTGCTGGTACGTGT | AGTTCCTCACCATCTCCTCGT | 57 |
| *waxy_13* | ATGTGTTTCCTCCTGGCTTG | TCCGGAGAAGTATGGGTTGT | 57 |
| *waxy_22* | TTCCCAGTGTAACGTCGTGG | AACACCGAACAGCAGGGATT | 57 |
| *waxy_25* | CGTGCACACGTCTTTTCTCTC | TACACGAACACAAGCCAGGAG | 57 |
| *waxy_26* | AGAAATACCGAGGCCTGGAC | GCCGATTAATCCACTGCGTA | 57 |
| *waxy_27* | CTGCGTGTTTGATGATCCAG | GGCATCAGAGCAGAGAAAGG | 57 |
| *waxy_29* | ACCCAAAAGTACCCACGACA | TCCAGGCCTCGGTATTTCTA | 57 |
| *waxy_30* | TCGTCTCCTGTGCTTCCTG | CTTGGCTTGTCGTCTGCAC | 57 |
| *waxy_31* | GCTGCTTGCTTGTGCTAGTG | GAAACAAACAGGGCCAAAGA | 57 |

5'-end contains part of the sequence of the second ORF of the full-length *Cin4*, and its 3'-end is a poly (A) tail of 9 bp; Its TSDs sequence is 5'-GCAACGCGATGGATAA-3'.

In total 53 *Cin4* insertion sites were identified from the B73 genome, of which 23 had complete TSDs structure (Table 3). Subsequently, the phylogenetic tree was established with 23 *Cin4* sequences, and The *Cin4* of *wx-Cin4*, *wx-Cin4-2* and *Cin4-2* were clustered into a single clade (Fig 4). The *Cin4-2* is a full-length *Cin4* with 7,199 bp, which has the complete structure of *Cin4* and is used as the reference sequence in this study. Furthermore, the sequences of *Cin4*

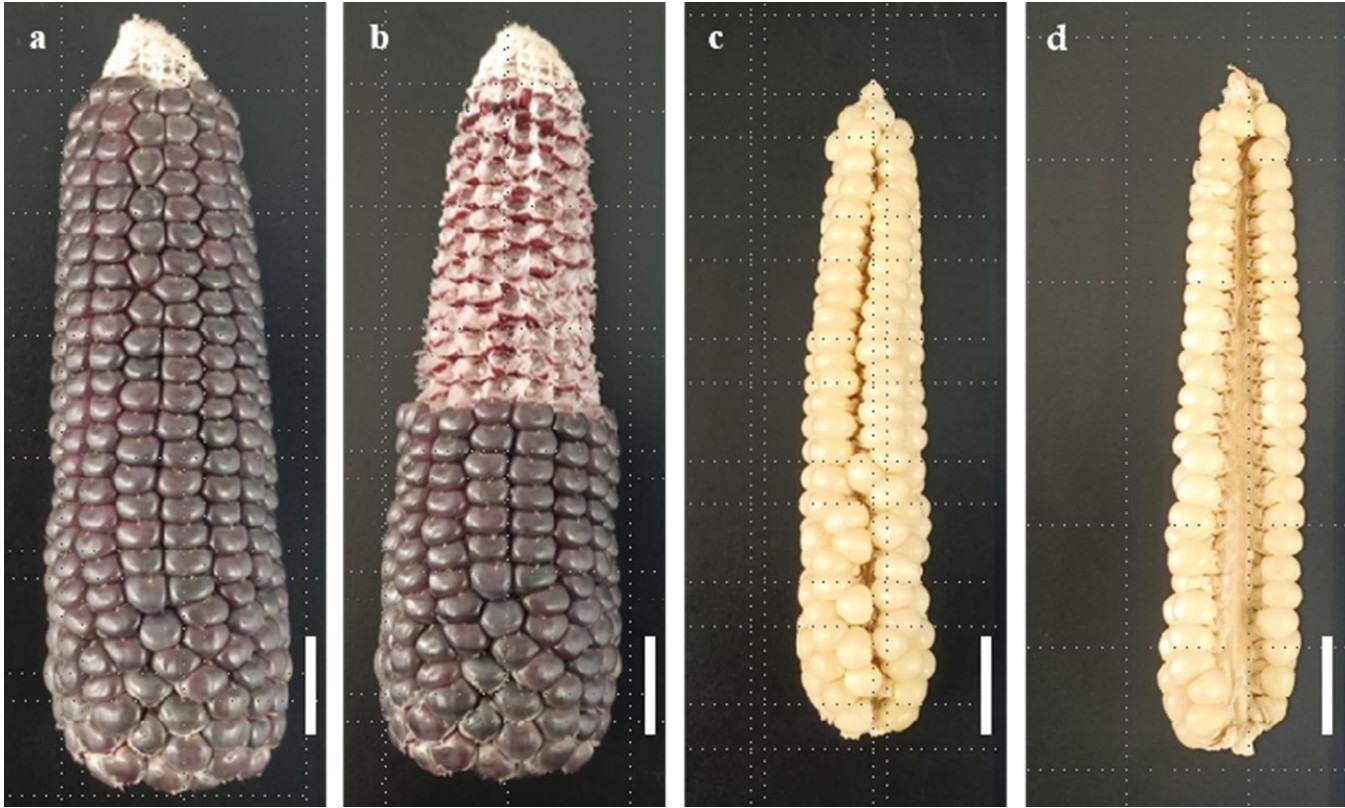

**Fig 2. Spike morphology of materials used in this study.** (a) is purple ears of Zinuoyumi; (b) is purple cob of Zinuoyumi; (c) is ear front of Myanmar Four-Row Wax; (d) is ear back of Myanmar Four-Row Wax. Bar = 2 cm.

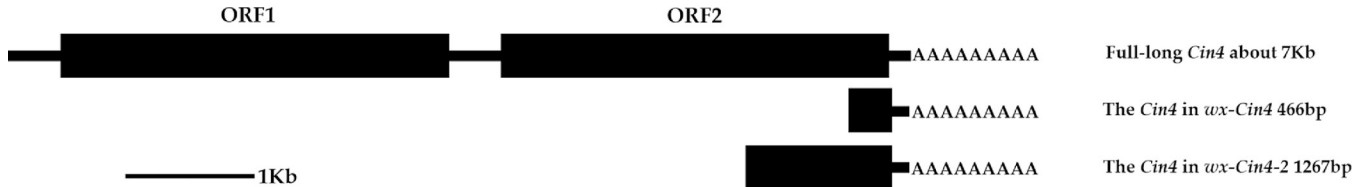

**Fig 3. Sequence and structure of the *Cin4* in *wx-Cin4-2*.** The full-length *Cin4* transposon is about 7 Kb in length and has two open reading frames (ORF) to encode the necessary enzymes for jumping in maize genome. The length of *Cin4* in *wx-Cin4-2* is 1,267 bp, which is a truncated transposon of *Cin4*. Both full-length and non-full-length *Cin4* have target site duplications (TSDs) sequence and poly (A) tail at 3' end.

in *wx-Cin4* and *wx-Cin4-2* were compared with *Cin4-2*, and the 5'-end sequences of both *Cin4* in *wx-Cin4* and *wx-Cin4-2* were found not to match *Cin4-2* completely, which are extra sequences of the *Cin4* (Fig 5). We found that the extra sequences come from the *Cin4* sequence itself, and there are overlapping sequences at the junction of the extra sequences with the target site sequence and with the *Cin4* sequence. The extra sequence of *Cin4* in *wx-Cin4-2* is 5'– ATAAGGGAGGTCT-3', the overlapping sequence at the junction with *wx-Cin4-2* is 5'-T- 3', and the overlapping sequence with TSD is 5'-ATAA-3' (Fig 5).

**Table 3. The non-LTR retrotransposon *Cin4* in B73 genome.**

| Name | Locus | TSDs | Size (bp) | *Cin4* Type | Poly(A) Size(bp) | Extra sequences Size(bp) |
|---|---|---|---|---|---|---|
| *Cin4-1* | Chr8.134,025,509–134,032,678 | 5'-AGTATTTACCAATGCA-3' | 7,170 | Full-length | 9 | 0 |
| *Cin4-2* | Chr4.178,591,377–178,598,575 | 5'-TACATCAGATTTGCT-3' | 7,199 | Full-length | 12 | 15 |
| *Cin4-4* | Chr10.141,035,476–141028,,296 | 5'-GAAGATGGAGAG-3' | 7,181 | Full-length | 8 | 17 |
| *Cin4-5* | Chr9.101,484,413–101,477,215 | 5'-TATATTGTGCTGCAGC-3' | 7,199 | Full-length | 7 | 26 |
| *Cin4-7* | Chr2.27,628,992–27632,,684 | 5'-CAAGGGA-3' | 3,693 | Truncated | 6 | 42 |
| *Cin4-8* | Chr5.194,690,425–194,697,588 | 5'-AGATATAGATCTAA-3' | 7,164 | Full-length | 7 | 190 |
| *Cin4-9* | Chr4.38,800,135–38,807,614 | 5'-AGTATTTACCAATGCA-3' | 7,480 | Full-length | 9 | 2378 |
| *Cin4-10* | Chr10.3,389,832–3,,394,724 | 5'-AATAGCTGACCAAA-3' | 4,893 | Truncated | 7 | 60 |
| *Cin4-14* | Chr3.214,892,867–214,894,478 | 5'-AAAGTTTC-3' | 1,612 | Truncated | 6 | 3 |
| *Cin4-15* | Chr10.73,221,653–73,223,683 | 5'-AGTTTCCT-3' | 2,031 | Truncated | 7 | 0 |
| *Cin4-17* | Chr1.40,093,443–40,095,169 | 5'-ATGTATTTAGC-3' | 1,727 | Truncated | 6 | 0 |
| *Cin4-18* | Chr1.40,093,443–40,095,169 | 5'-ATGTATTTAGC-3' | 1,727 | Truncated | 6 | 0 |
| *Cin4-20* | Chr2.200,214,986–200,217,304 | 5'-ATATCCTCATATC-3' | 2,319 | Truncated | 12 | 0 |
| *Cin4-22* | Chr1.245654,,252–245652,,689 | 5'-GAGATATTTTAGATGC-3' | 1,564 | Truncated | 6 | 0 |
| *Cin4-24* | Chr3.213,671,863–213673,,599 | 5'-AGAGAGTG-3' | 1,737 | Truncated | 9 | 434 |
| *Cin4-25* | Chr7.172,960159–172,,961,493 | 5'-AACAATATC-3' | 1,335 | Truncated | 7 | 72 |
| *Cin4-30* | Chr5.15,606,409–15,605,239 | 5'-GATATACATCC-3' | 1,171 | Truncated | 8 | 0 |
| *Cin4-31* | Chr5.5,006,082–5,004,369 | 5'-GGAAAAGGCGGAT-3' | 1,714 | Truncated | 6 | 5 |
| *Cin4-39* | Chr1.245,536,944–245,537,771 | 5'-AAGTTTTTTTAGA-3' | 828 | Truncated | 6 | 19 |
| *Cin4-44* | Chr10.126,585,700–126,584,697 | 5'-AAGAACCATGGATGATT-3' | 1,004 | Truncated | 7 | 6 |
| *Cin4-46* | Chr9.23,067,142–23,066,186 | 5'-AGATTGAGCACTAGT-3' | 957 | Truncated | 7 | 27 |
| *Cin4-49* | Chr2.156,584,140–156,583,308 | 5'-GATATCATTTTTGGTTTT-3' | 833 | Truncated | 9 | 17 |
| *Cin4-51* | Chr10.104,752,134–104,754,059 | 5'-AAGATAAC-3' | 1,926 | Truncated | 7 | 1216 |
| *wx-Cin4* | *waxy* | 5'-AAGAGTTGCAGTCTTCG-3' | 466 | Truncated | 9 | 64 |
| *wx-Cin4-2* | *waxy* | 5'-AGCAACGCGATGGATAA-3' | 1,267 | Truncated | 9 | 9 |

The *wx-Cin4-2* has been identified only in Zinuoyumi. Two *waxy* alleles caused by *Cin4* transposon insertion were identified in Zinuoyumi, suggesting that maybe active *Cin4* transposons exist in the genome of Zinuoyumi.

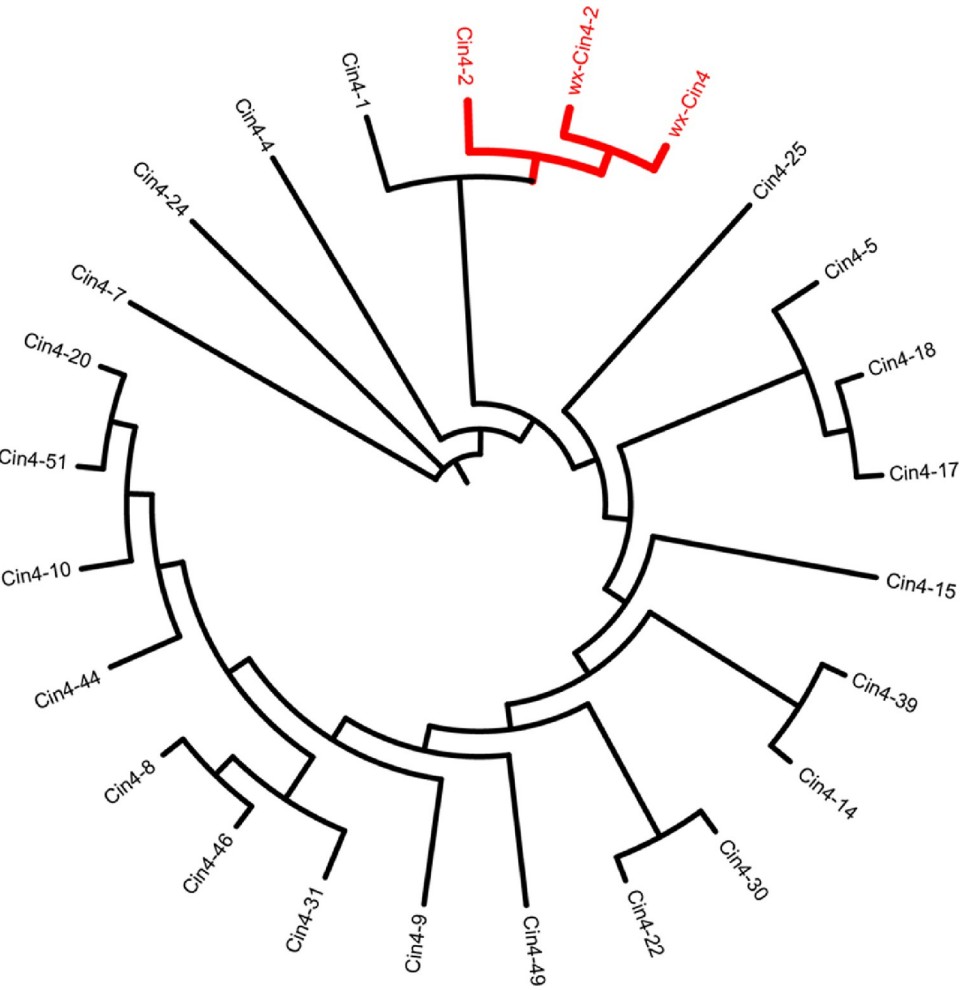

**Fig 4. The NG tree of *Cin4* from *wx-Cin4*, *wx-Cin4-2* and 23 *Cin4* from B73 genome.** The *Cin4* from *wx-Cin4* and *wx-Cin4-2* is divided on the same branch as the full-length *Cin4 Cin4-2*.

### Myanmar Four-Row Wax

Four-Row Wax has long been regarded as one of the specific waxy maize landraces in the Yunnan area because its rows number of ears is four (Fig 2). Four-Row Wax was first found in the Dai nationality residential area of Menghai, Yunnan Province, and it is also distributed in other areas. Yunnan provincial repository for corps germplasm now stores Four-Row Wax collected from the Menghai, Menglian, Yingjiang and Namhkan areas of the China-Myanmar border, and the Four-Row Wax from Namhkan was named Myanmar Four-Row Wax by us.

It was previously reported that the *waxy* gene of Four-Row Wax is *wx-D10* [16, 17, 31]. However, our study found that Myanmar Four-Row Wax has three different alleles, including *wx-D10*, *wx-Reina* and *wx-D11* which is a new mutation found in this study. Through sequence alignment with the wild-type *waxy* gene, it was found that the *wx-D11* is a 1,082 bp deletion mutation from exon 11 to exon 14 (Fig 1), and replaced by a 72 bp filler sequence. The filler sequence has an 11 bp TSDs structure like to transposon of the genome, and the sequence is 5′ -TGGTACGTGTG-3′. The filler sequence between TSDs can be divided into three parts: Part I is an 18 bp forward sequence from exon 10 of *waxy* gene; Part II is a 17 bp unknown source sequence; Part III is a 37 bp reverse complementary sequence from exon 14

of *waxy* gene (Fig 6). In addition, we noted that the mutation site of *wx-D11* was consistent with *wx-Reina*, but the TSDs sequence was not the same as *wx-Reina*.

Specific markers for *wx-Cin4-2* and *wx-D11* were further developed in this study, and agarose gel showed amplicons from Zinuoyumi and Myanmar Four-Row Wax (Fig 7).

## Discussion

### Yunnan and its surrounding areas are the secondary origin center of waxy maize

Vavilov's theory on the origin center of crops [32]: The primary origin center is the primary origin center. When the crops spread to a certain range, they will form a new recessive gene-

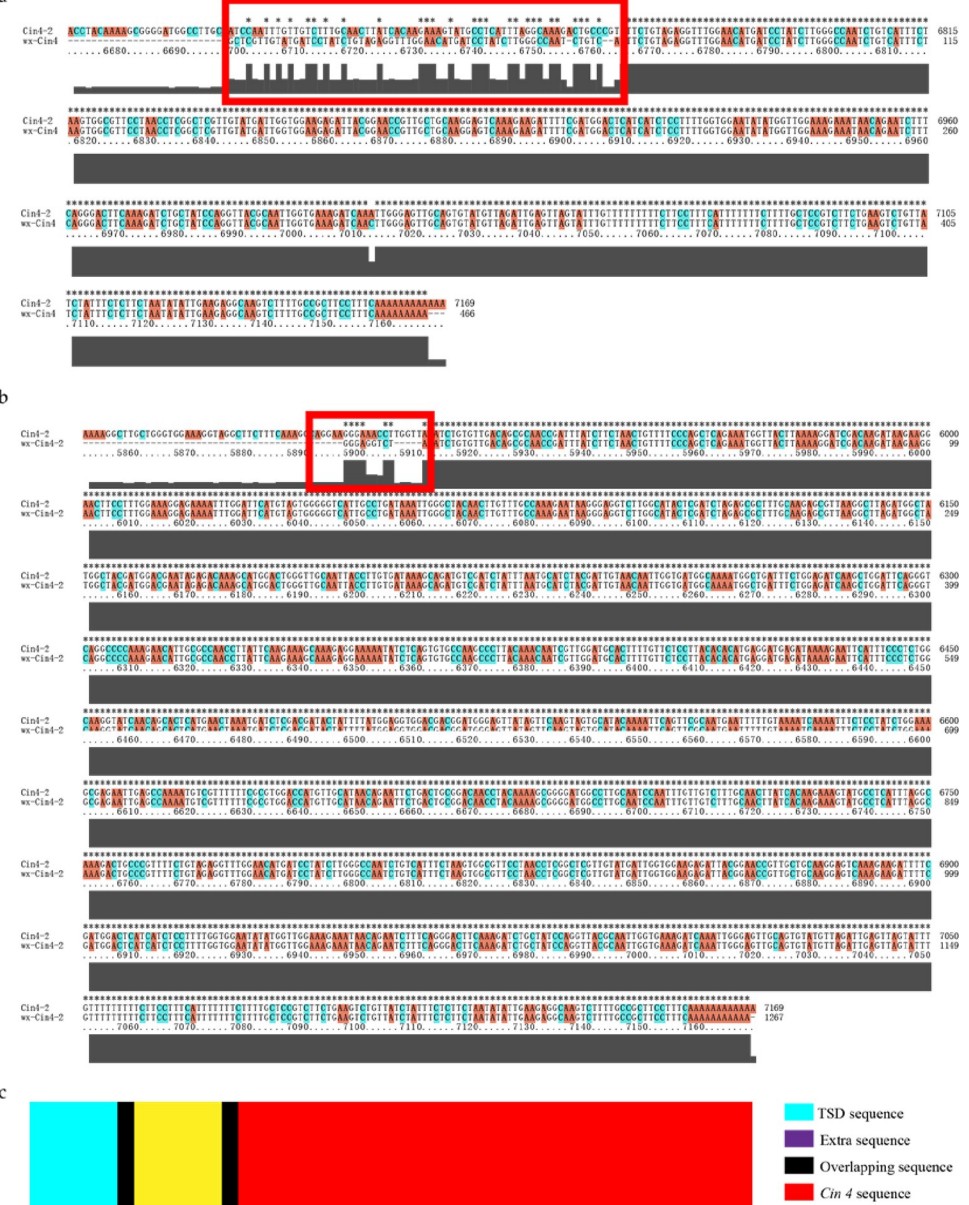

**Fig 5. The extra sequence of *Cin4*.** (a) Sequence alignment of *wx-Cin4* and *Cin4-2*; (b) Sequence alignment of *wx-Cin4-2* and *Cin4-2*; (c) the structure of extra sequence. There were overlapping sequences between the extra sequences, the target sequence and *Cin4* sequence. The red box indicated that the 5' end sequences of *wx-Cin4-2* and *wx-Cin4* were found not to match *Cin4-2*, which are extra sequences of *Cin4*.

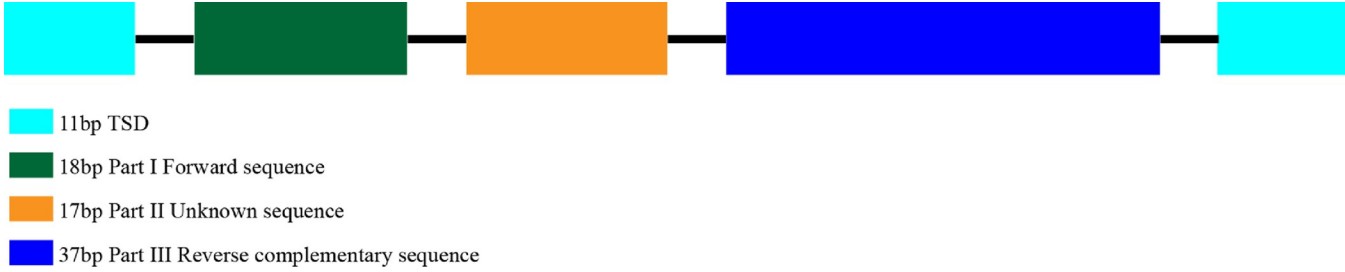

**Fig 6. The filler sequence structure of *wx-D11*.** The filled sequence has 11 bp TSDs structure `5'-TGGTACGTGTG -3'`. The filler sequence between TSDs can be divided into three parts: Part I is an 18 bp forward sequence from exon 10 of *waxy* gene `5'-CGCCGCCCGCCACCCGGC-3'`; Part II is a 17 bp unknown source sequence `5'-CACCACAAGCACACTAT-3'`; Part III is a 37 bp reverse complementary sequence from exon 14 of *waxy* gene `5'-TATATTACACTAGCACAAGCAAGCAGCTACACATACT-3'`.

controlled diversification area, namely the secondary origin center or secondary gene center, due to the self-crossing and natural isolation of the crops. The secondary origin also has four characteristics: (1) no wild ancestors; (2) there are new specific types; (3) there is a lot of variation; (4) there are a lot of recessive genes.

So far, it can be seen that there is no wild ancestor of maize in the Yunnan region; however, a large number of waxy maize resources were accumulated in this area with recessive waxy genes. Some of them are unique to this region, and these materials have special morphological characteristics; abundant recessive gene variation was observed from the *waxy* site. In addition, we suggest that the phenomenon of multiple alleles is a major molecular characteristic of ancient landraces, and the accumulation of *waxy* alleles is caused by selection pressure in Yunnan waxy landraces.

## The influence of Chinese waxy food culture on crop selection

Maize originated in the Americas continent and was introduced to China about 400–500 years ago [1, 33]. It is generally believed in the academic community that waxy maize is formed by selection after gene mutations in common maize [34]. Waxy maize has been planted in China for over 200 years [6]. Maize undergoes genetic mutations within one to two hundred years after being introduced into China. These mutations were noticed by Zhuang, Dong, Buyi, Dai and other ethnic minorities in southwest China who lived in the "waxy rice cultivation area" or "waxy rice cultural circle" or "waxy food cultural circle", and these mutations were selected and

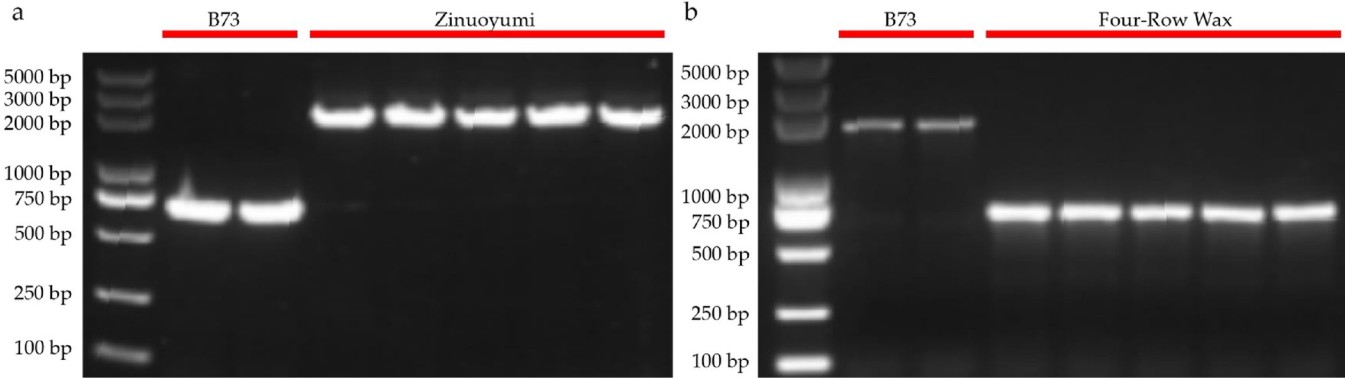

**Fig 7. The 1% agarose gel electrophoresis results of the simplified the *wx-Cin4-2* and the *wx-D11*.** (a), the *wx-Cin4-2* from Zinuoyumi; (b), the *wx-D11* from Myanmar Four-Row Wax.

retained. These ethnic minorities were developed from their ancestors, the Baiyue people, during the Ming and Qing dynasties. The Baiyue people were the earliest rice cultivators in China, as well as the earliest waxy rice cultivators and eaters. These ethnic minorities have learned and borrowed from their ancestors' experience and knowledge in the selection of waxy rice for the selection of waxy maize [35, 36]. The waxy mutant of maize, waxy maize, can be preserved and accumulated, forming hundreds of landraces or germplasm resources. Therefore, the waxy food culture and knowledge of ethnic minorities in southwest China, especially in Yunnan and surrounding areas, is the main driving force for the formation of waxy maize in China.

## Transposon activity is the main driving force for the formation of *waxy* allele diversity

Transposon insertion is a major form of *waxy* gene mutation in maize. Transposons can be divided into DNA transposons and RNA transposons based on the difference in the transposon jumping medium within the genome [37]. RNA transposons can further divide into long terminal repeat (LTR) retrotransposon and non-LTR retrotransposon. Compared with non-LTR retrotransposon, LTR retrotransposon has long forward repeat LTR at 5'-end and 3'-end. Non-LTR retrotransposons usually have a 3'-poly (A). Whereas LTR retrotransposon can be further divided into Ty1-*Copia* and Ty3-*Gypsy* according to the order of their domains [38]. Miniature inverted-repeat transposable element (MITE) is a kind of DNA transposon which is mainly non-self-transposable [39, 40]. The insertion of different types of transposons represents the structural characteristics of different types of *waxy* alleles. Our previous research found that, the *waxy* gene mutation type of waxy maize in Yunnan province involves the above-mentioned transposons (Table 4). The mutations of *wx-Cin4-2* were also caused by transposon activity. This further indicates that the activity of transposon is the main cause of the formation of the *waxy* allele.

## Formation of an extra sequence of *Cin4* and its biological role

Insertion of the non-LTR retrotransposon into chromosomal DNA is reported to be initiated by a mechanism called target-primed reverse transcription (TPRT), in which the target DNA was cleaved to generate a free hydroxyl (OH). This hydroxyl acts as a primer for reverse transcription using retrotransposon RNA as a template [41, 42]. However, more details of the second strand synthesis are not very clear [43]. In some reports, the following step of transposition was the cleavage of the second strand. The newly synthesized cDNA jumps from the retrotransposon RNA to the second strand at the target site. The second-strand cleavage creates a primer for second-strand synthesis. The primer was a microhomology (MH) sequence between the 5'-end of non-LTR retrotransposon and its target site [42, 44]. The formation of an extra sequence can change the terminal sequence of transposons, which is conducive to the formation of MH sequences and promotes template switching (Fig 8).

**Table 4. The copy number of transposon found in different waxy alleles in the B73 genome.**

| Allele | Transposon family | Transposon classification | Transposon length (bp) | TSD sequence (5'-3') | Copy number |
|---|---|---|---|---|---|
| *wx-124* | *124* | MITE | 116 | CTCGTGCTA | ~13 |
| *wx-Reina* | *Reina* | LTR retrotransposon (Ty3-*Gypsy*) | 5,412 | GTGTG | ~4 |
| *wx-xuanwei* | *xuanwei* | LTR retrotransposon (Ty1-*Copia*) | 4,893 | CTGCC | ~2 |
| *wx-PIF/Harbinger* | *PIF/Harbinger* | MITE | 301 | GGCTTTG | >500 |
| *wx-hAT* | *hAT* | MITE | 560 | GCTTGCTGG | >500 |
| *wx-Elote2* | *Elote2* | LTR-like | 6,560 | TTCAC | >500 |
| *wx-Cin4* | *Cin4* | Non-LTR retrotransposon (*LINE*) | 466 | AAGAGTTGCAGTCTTCG | ~53 |
| *wx-Cin4-2* | *Cin4* | Non-LTR retrotransposon (*LINE*) | 1,267 | AGCAACGCGATGGATAA | ~53 |

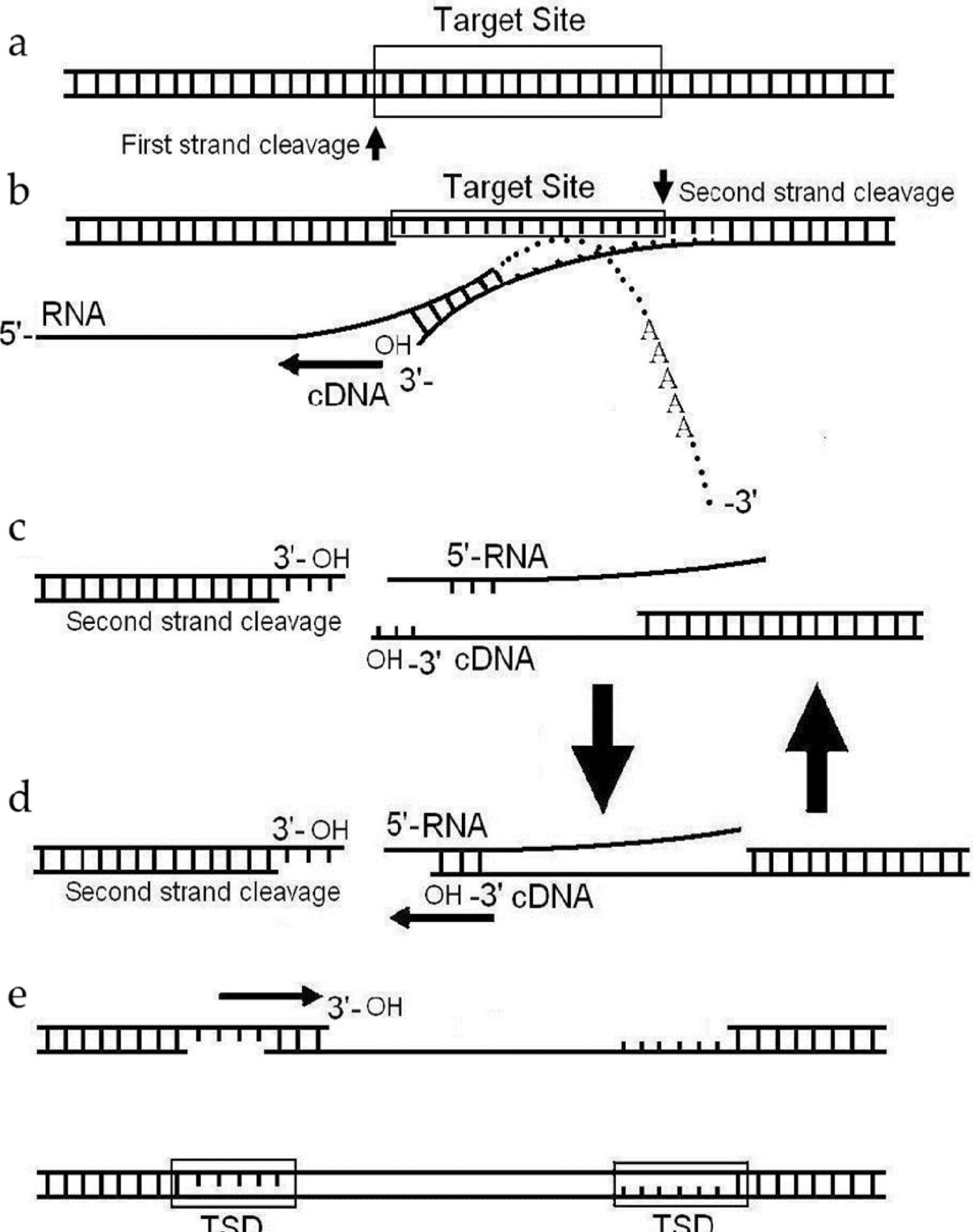

**Fig 8. The possible process of *Cin4* jump.** (a) The first strand was cleaved at the transposition target site, producing a free hydroxyl (OH) at the nick. (b) The cDNA synthesis used the free hydroxyl (OH) as a primer following template RNA, and the second strand was cleaved at transposition target site. (c) Template RNA leaved from the cDNA during the synthesis of cDNA. After the 3' end of cDNA annealed to a new site of the template RNA, cDNA synthesis restarted. (d) The cleavage of the second strand annealed to cDNA, the second strand synthesis started following newly cDNA. (e) A copy of element was integrated at a new genomic location and was flanked by target site duplications (TSDs). The extra sequence is formed in process (c) and remains in *Cin4* after transposition.

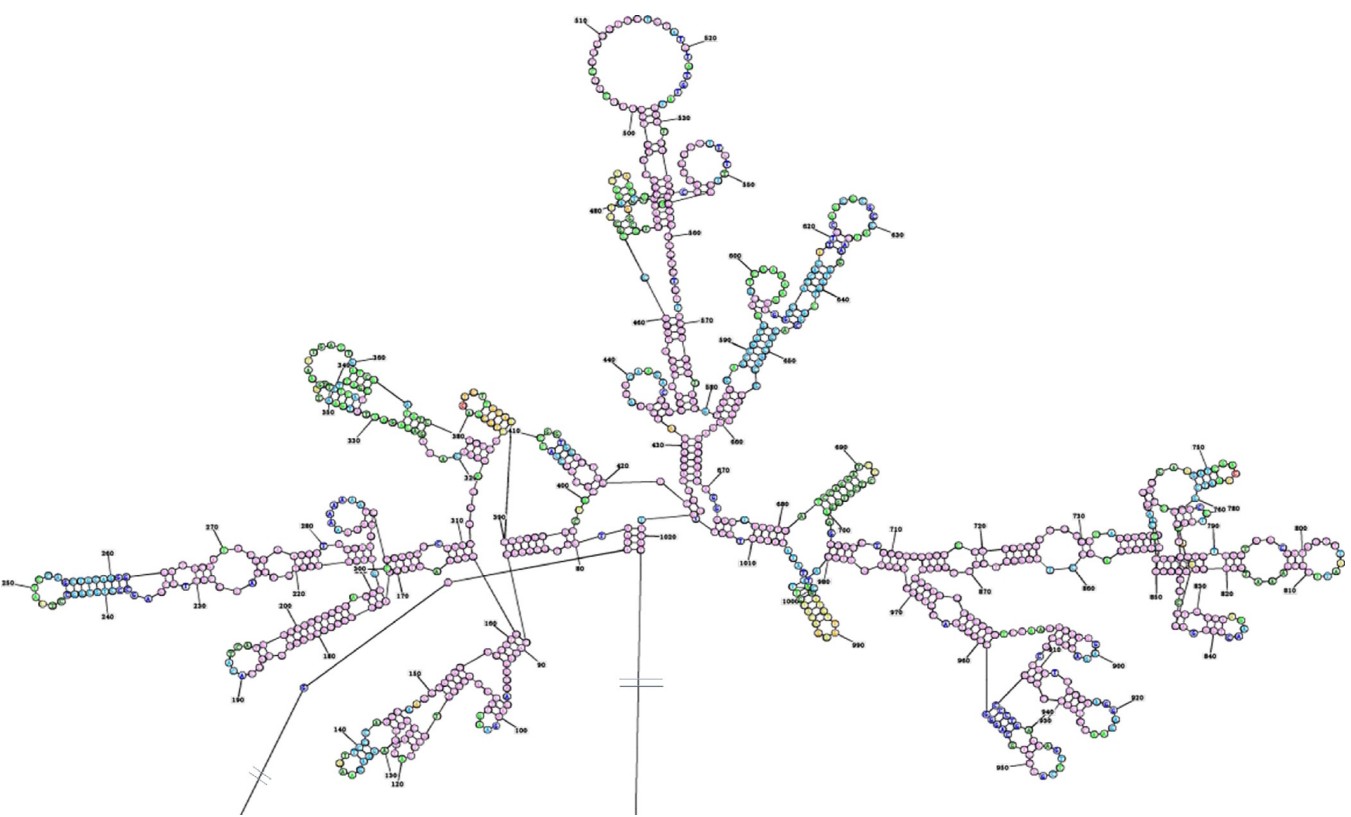

**Fig 9. The secondary structure of the *wx-D11* deletion sequence (1,082 bp).** The deletion sequence can constitute a stem-loop structure with arms and rings.

## Formation of *wx-D11* mutation site

Previous studies have found that *wx-B*, *wx-BI*, *wx-B6*, and *wx-C4* are mutations due to *waxy* gene sequence deletions from exons 1 to 7, and the deletion sequences are replaced by filling sequences [45]. These structures are the result of more than one molecular mechanism in the genome. One possibility is that the sequence of the deletion site can form a stable secondary structure. The secondary structure cannot be effectively opened during DNA replication. During the DNA replication process, the absence of replication results in the deletion of the DNA sequence at this site. The deletion of sequences triggers the repair mechanism of the genome, and the repair process introduces filling sequences [45]. We analyzed the secondary structure of the deletion sequence of *wx-D11* and found that this site could form a typical secondary structure (Fig 9). The other is that, DNA transposition can carry some of its flanking sequences jump to other parts of the genome; sometimes the transposition could induce double-strand DNA breakage, and then the homologous recombination repair mechanism of chromosome would be triggered to repair the DNA in this position, the repair may not be so perfect, caused the formation of filler sequence at the break sites [46].

The gene sequence in this article has been submitted to the genebank database and the sequence number *wx-Cin4-2* (OR161363) *wx-D11* (OR161364) has been obtained.

## Supporting information

**S1 Raw images.**
(ZIP)

## Author Contributions

**Data curation:** Tingting Sun, Xiaoyang Wu.

**Funding acquisition:** Xiaoyang Wu.

**Writing – original draft:** Tingting Sun, Xiaoyang Wu.

**Writing – review & editing:** Tingting Sun, Xiaoyang Wu.

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
