## [Decision Letter · Decision Letter 0]

13 Jun 2023

PONE-D-23-09518Structure Characteristics of Mutation Sites in Two Waxy Alleles from Yunnan Waxy LandracesPLOS ONE

Dear Dr. Wu,

Thank you for submitting your manuscript to PLOS ONE. After careful consideration, we feel that it has merit but does not fully meet PLOS ONE’s publication criteria as it currently stands. Therefore, we invite you to submit a revised version of the manuscript that addresses the points raised during the review process.

We look forward to receiving your revised manuscript.

Kind regards,

Diaa Abd El-Moneim

Academic Editor

PLOS ONE

Journal Requirements:

https://link.springer.com/article/10.1007/s10722-019-00763-z?code=2dc26c94-5697-460e-9d56-b1f22a4601f3&error=cookies_not_supported 

https://repository.cimmyt.org/xmlui/bitstream/handle/10883/3264/97999.pdf?isAllowed=y&sequence=1

In your revision ensure you cite all your sources (including your own works), and quote or rephrase any duplicated text outside the methods section. Further consideration is dependent on these concerns being addressed.

This work was supported by the National Natural Science Foundation of China (32060459)

7. We note that Figure 2 in your submission contain map images which may be copyrighted. All PLOS content is published under the Creative Commons Attribution License (CC BY 4.0), which means that the manuscript, images, and Supporting Information files will be freely available online, and any third party is permitted to access, download, copy, distribute, and use these materials in any way, even commercially, with proper attribution. For these reasons, we cannot publish previously copyrighted maps or satellite images created using proprietary data, such as Google software (Google Maps, Street View, and Earth). For more information, see our copyright guidelines: http://journals.plos.org/plosone/s/licenses-and-copyright.

8. Please include your tables as part of your main manuscript and remove the individual files. Please note that supplementary tables (should remain/ be uploaded) as separate "supporting information" files

Reviewers' comments:

Reviewer's Responses to Questions

**Comments to the Author**

1. Is the manuscript technically sound, and do the data support the conclusions?

Reviewer #1: Yes

Reviewer #2: Yes

Reviewer #3: Yes

2. Has the statistical analysis been performed appropriately and rigorously? 

Reviewer #1: Yes

Reviewer #2: Yes

Reviewer #3: N/A

3. Have the authors made all data underlying the findings in their manuscript fully available?

Reviewer #1: Yes

Reviewer #2: Yes

Reviewer #3: Yes

4. Is the manuscript presented in an intelligible fashion and written in standard English?

Reviewer #1: Yes

Reviewer #2: Yes

Reviewer #3: No

5. Review Comments to the Author

Reviewer #1: Dear Prof. Dr. Editor of the PLOS ONE Journal,

I write you regarding Manuscript ID: PONE-D-23-09518 entitled "Structure Characteristics of Mutation Sites in Two Waxy Alleles from Yunnan Waxy Landraces" which was submitted to the PLOS ONE Journal.

In this manuscript, the authors studied the Structure Characteristics of Mutation Sites in Two Waxy Alleles from Yunnan Waxy Landraces.

This work was well done and appropriate. The manuscript is suitable for publication in the International Journal of Molecular Sciences.

I have gone through this work. My decision is accepted with minor revisions for this work. The reason for that is as follows:

The manuscript deals with " Structure Characteristics of Mutation Sites in Two Waxy Alleles from Yunnan Waxy Landraces".

First: Title: It should change to the following:

1) Structure Characteristics of Mutation Sites in Two Waxy Alleles of Waxy Maize (Zea mays L.) Landraces from Yunnan

Second Abstract, keywords and Introduction:

2) has some minor corrections as in the attached file.

Third: The other parts

3) has some minor corrections as in the attached file.

References

4) has some minor corrections as in the attached file.

Thank you for suggesting me as a reviewer for this manuscript.

with best regards

Reviewer #2: Paper entitled" Structure Characteristics of Mutation Sites in Two Waxy Alleles from Yunnan Waxy

Landraces" was in the scope of journal and writien in scientific language with good English editing. Study the waxy genes is very important for Maize production and increasing the yield productivity. I recommend publication of this manuscript after minor revision.

Introduction. Need more explanation in the role of waxy genes in productivity of Maize and other phenotypic characters.

Results. You can characterize the Waxy genes in genbank database and compare it with the genes present in the data base

Discussion. Needs more explanation for the genes and compare it with previous study

Reviewer #3: The manuscript entitled “Structure Characteristics of Mutation Sites in Two Waxy Alleles from Yunnan Waxy Landraces", is well-written with an extensive discussion and addresses the main objectives of the study. It provides valuable insights into the Structural Characteristics of Waxy genes. In my review, I carefully and thoroughly assessed the content of the report, also, I have identified several key points that I believe are important to highlight. However, there are areas where improvements can be made, that may benefit from further clarification or revision of the article (Manuscript ID: PONE-D-23-09518- PLOS ONE). The suggestions below are:

Line#93 (Similarly,for line#132), Author stated that “two different mutation sites and 30bp deletion in exon 10” in wx-Cin4-2 of Zinuoyumi. However, it not clearly mentioned or work done anywhere in the entire manuscript, with reference to what, author stated two mutation sites and deletion30bp? What kind of mutation is (sense or nonsense, homo or heterozygous)? What is the consequence of these mutations, although author has partly explained in the discussion section, but need to explain in detail. Because "sense" or "nonsense" describes the impact of a mutation on protein function, while "homozygous" or "heterozygous" refers to the genetic state of an individual with respect to the mutation. These terms are distinct and address different aspects of genetic mutations.

Additionally, as author has shown specificity on agarose gel, I don’t see sanger sequencing confirmed mutation using chromatogram. That will help to assess mutation type and other.

Line#119, Author stated, “The wx-Cin4-2 has been identified only in Zinuoyumi. Two waxy alleles caused by Cin4 transposon insertion were identified in Zinuoyumi, suggesting that maybe active Cin4 transposons exist in genome of Zinuoyumi. However, what is the advantage of having this mutation in Zinuoyumi is not clear, whether it has any survival advantage association.

Line#146-147

Yunnan and its surrounding areas are the secondary origin center of waxy maize Vavilov’s theory on origin center of crops: Why half is bold, author need to continue and reframe sentence without colon sign”:” and bold.

Line#146-159, the reference is missing in the entire section.

The entire discussion in the manuscript is so lengthy and there is lot of irrelevant matter and can be rearranged from discussion part to the introduction section for example Line #178-193 can be added to the introduction part but just brief little in discussion rather lengthy.

Line#44, why author has started sentence with “AND” as it can be use and to connect two words, phrases, clauses or prefixes together, Author needs to correct this.

Line #46 and 48, Author needs to reframe the sentence “ The known alleles include wx-D7, wx-D10, wx-Cin4, wx-124, wx-Reina, wx-Xuanwei, wx PIF/Harbinger, wx-hAT, and wx-Elote2 [13-16]; wx-D7 and wx-D10 “were mutations” caused by sequence deletion in waxy gene [17, 18]; wx-Cin4, wx-124, wx-Reina, wx-Xuanwei, wx-PIF/Harbinger, wx-hAT, and wx-Elote2 “are mutations” caused by transposon insertion in waxy gene [13-16]; wx-D10 and wx-Reina are widely distributed waxy alleles in waxy maize landraces of Yunnan area.

Instead, author should write “mutations were” caused by sequence deletion in waxy gene [17, 18], and similarly, , wx-hAT, and wx-Elote2 “having mutations.”

Similarly, author stated that, previous studies have found that wx-B, wx-BI, wx-B6, and wx-C4 “are” mutations due to waxy gene sequence deletions from exons 1 to 7, and the deletion sequences are replaced by filling sequences. Author should reframe “are” instead use “have”.

Line# 54-55, Author has stated that “The identification of waxy allele is helpful to the protection and utilization of these waxy landraces” However, it seems incomplete sentence, what author want to state here, not clear. Similarly, Author stated that, “ large number of Yunnan waxy maize landraces still contain unknown waxy alleles that need to be identified, I am wondering how author knows about this is not clear, whether it is functional characterization terms or some other mean and promoter has not been assigned in figure 1 although CCAAT and TATAA box has being mentioned and The major thing I don’t see is the functional association with these mutations is missing.

6. PLOS authors have the option to publish the peer review history of their article (what does this mean?). If published, this will include your full peer review and any attached files.

Reviewer #1: **Yes: **Khaled F. M. Salem

Reviewer #2: No

Reviewer #3: **Yes: **MUDASIR RASHID

---

## [Author Response · Author response to Decision Letter 0]

26 Jul 2023

Comments to the Author

1. Is the manuscript technically sound, and do the data support the conclusions?

Reviewer #1: Yes

Reviewer #2: Yes

Reviewer #3: Yes

2. Has the statistical analysis been performed appropriately and rigorously?

Reviewer #1: Yes

Reviewer #2: Yes

Reviewer #3: N/A

3. Have the authors made all data underlying the findings in their manuscript fully available?

Reviewer #1: Yes

Reviewer #2: Yes

Reviewer #3: Yes

4. Is the manuscript presented in an intelligible fashion and written in standard English?

Reviewer #1: Yes

Reviewer #2: Yes

Reviewer #3: No

5. Review Comments to the Author

Reviewer #1: Dear Prof. Dr. Editor of the PLOS ONE Journal,

I write you regarding Manuscript ID: PONE-D-23-09518 entitled "Structure Characteristics of Mutation Sites in Two Waxy Alleles from Yunnan Waxy Landraces" which was submitted to the PLOS ONE Journal.

In this manuscript, the authors studied the Structure Characteristics of Mutation Sites in Two Waxy Alleles from Yunnan Waxy Landraces.

This work was well done and appropriate. The manuscript is suitable for publication in the International Journal of Molecular Sciences.

I have gone through this work. My decision is accepted with minor revisions for this work. The reason for that is as follows:

The manuscript deals with " Structure Characteristics of Mutation Sites in Two Waxy Alleles from Yunnan Waxy Landraces".

First: Title: It should change to the following:

1) Structure Characteristics of Mutation Sites in Two Waxy Alleles of Waxy Maize (Zea mays L.) Landraces from Yunnan

Response:

Thank you for your constructive comments.

We have revised the title of this manuscript.

Second Abstract, keywords and Introduction:

2) has some minor corrections as in the attached file.

Response:

Thank you for your constructive comments.

Third: The other parts

3) has some minor corrections as in the attached file.

Response:

Thank you for your constructive comments.

References

4) has some minor corrections as in the attached file.

Response:

Thank you for your constructive comments.

Thank you for suggesting me as a reviewer for this manuscript.

with best regards

Reviewer #2: Paper entitled" Structure Characteristics of Mutation Sites in Two Waxy Alleles from Yunnan Waxy

Landraces" was in the scope of journal and writien in scientific language with good English editing. Study the waxy genes is very important for Maize production and increasing the yield productivity. I recommend publication of this manuscript after minor revision.

Introduction. Need more explanation in the role of waxy genes in productivity of Maize and other phenotypic characters.

Response:

Thank you for your constructive comments.

We added the role of the waxy gene in the starch synthesis pathway in the introduction.

Results. You can characterize the Waxy genes in genbank database and compare it with the genes present in the data base

Response:

Thank you for your constructive comments.

We compared the characteristics of different types of waxy alleles in Table 4.

Discussion. Needs more explanation for the genes and compare it with previous study

Response:

Thank you for your constructive comments.

The “Transposon activity is the main driving force for the formation of waxy allele diversity” part in discussed of this article was compared with our previous work. 

Reviewer #3: The manuscript entitled “Structure Characteristics of Mutation Sites in Two Waxy Alleles from Yunnan Waxy Landraces", is well-written with an extensive discussion and addresses the main objectives of the study. It provides valuable insights into the Structural Characteristics of Waxy genes. In my review, I carefully and thoroughly assessed the content of the report, also, I have identified several key points that I believe are important to highlight. However, there are areas where improvements can be made, that may benefit from further clarification or revision of the article (Manuscript ID: PONE-D-23-09518- PLOS ONE). The suggestions below are:

Line#93 (Similarly, for line#132), Author stated that “two different mutation sites and 30bp deletion in exon 10” in wx-Cin4-2 of Zinuoyumi. However, it not clearly mentioned or work done anywhere in the entire manuscript, with reference to what, author stated two mutation sites and deletion30bp? What kind of mutation is (sense or nonsense, homo or heterozygous)? What is the consequence of these mutations, although author has partly explained in the discussion section, but need to explain in detail. Because "sense" or "nonsense" describes the impact of a mutation on protein function, while "homozygous" or "heterozygous" refers to the genetic state of an individual with respect to the mutation. These terms are distinct and address different aspects of genetic mutations.

Additionally, as author has shown specificity on agarose gel, I don’t see sanger sequencing confirmed mutation using chromatogram. That will help to assess mutation type and other.

Response:

Thank you for your constructive comments.

The 30bp deletion in exon 10 is a type of mutation found in previous studies. Besides the 30bp deletion in exon 10, wx-Cin4-2 also has the transposable element Cin4 insertion mutation.

Line#119, Author stated, “The wx-Cin4-2 has been identified only in Zinuoyumi. Two waxy alleles caused by Cin4 transposon insertion were identified in Zinuoyumi, suggesting that maybe active Cin4 transposons exist in genome of Zinuoyumi. However, what is the advantage of having this mutation in Zinuoyumi is not clear, whether it has any survival advantage association.

Response:

Thank you for your constructive comments.

There are still active Transposable element in the genome of Landrace of waxy maize in Yunnan, perhaps Cin4 in the genome of Zinuoyumi is one of them.

Line#146-147

Yunnan and its surrounding areas are the secondary origin center of waxy maize Vavilov’s theory on origin center of crops: Why half is bold, author need to continue and reframe sentence without colon sign”:” and bold.

Response:

Thank you for your constructive comments.

We have modified the font.

Line#146-159, the reference is missing in the entire section.

Response:

Thank you for your constructive comments.

We have added references.

The entire discussion in the manuscript is so lengthy and there is lot of irrelevant matter and can be rearranged from discussion part to the introduction section for example Line #178-193 can be added to the introduction part but just brief little in discussion rather lengthy.

Line#44, why author has started sentence with “AND” as it can be use and to connect two words, phrases, clauses or prefixes together, Author needs to correct this.

Response:

Thank you for your constructive comments.

We have modified this sentence.

Line #46 and 48, Author needs to reframe the sentence “ The known alleles include wx-D7, wx-D10, wx-Cin4, wx-124, wx-Reina, wx-Xuanwei, wx PIF/Harbinger, wx-hAT, and wx-Elote2 [13-16]; wx-D7 and wx-D10 “were mutations” caused by sequence deletion in waxy gene [17, 18]; wx-Cin4, wx-124, wx-Reina, wx-Xuanwei, wx-PIF/Harbinger, wx-hAT, and wx-Elote2 “are mutations” caused by transposon insertion in waxy gene [13-16]; wx-D10 and wx-Reina are widely distributed waxy alleles in waxy maize landraces of Yunnan area.

Response:

Thank you for your constructive comments.

We have modified this sentence.

Instead, author should write “mutations were” caused by sequence deletion in waxy gene [17, 18], and similarly, , wx-hAT, and wx-Elote2 “having mutations.”

Response:

Thank you for your constructive comments.

We have modified this sentence.

Similarly, author stated that, previous studies have found that wx-B, wx-BI, wx-B6, and wx-C4 “are” mutations due to waxy gene sequence deletions from exons 1 to 7, and the deletion sequences are replaced by filling sequences. Author should reframe “are” instead use “have”.

Response:

Thank you for your constructive comments.

We have modified this sentence.

Line# 54-55, Author has stated that “The identification of waxy allele is helpful to the protection and utilization of these waxy landraces” However, it seems incomplete sentence, what author want to state here, not clear. Similarly, Author stated that, “large number of Yunnan waxy maize landraces still contain unknown waxy alleles that need to be identified, I am wondering how author knows about this is not clear, whether it is functional characterization terms or some other mean and promoter has not been assigned in figure 1 although CCAAT and TATAA box has being mentioned and The major thing I don’t see is the functional association with these mutations is missing.

Response:

Thank you for your constructive comments.

We have modified this sentence.

We have modified this sentence.

The CCAAT and TATAA box represent the location of gene transcription initiation.

6. PLOS authors have the option to publish the peer review history of their article (what does this mean?). If published, this will include your full peer review and any attached files.

Do you want your identity to be public for this peer review? For information about this choice, including consent withdrawal, please see our Privacy Policy.

Reviewer #1: Yes: Khaled F. M. Salem

Reviewer #2: No

Reviewer #3: Yes: MUDASIR RASHID

---

## [Decision Letter · Decision Letter 1]

23 Aug 2023

Structure characteristics of mutation sites in two waxy alleles from Yunnan waxy maize (Zea mays L. var. certaina Kulesh) landraces

PONE-D-23-09518R1

Dear Dr. Wu,

We’re pleased to inform you that your manuscript has been judged scientifically suitable for publication and will be formally accepted for publication once it meets all outstanding technical requirements.

Kind regards,

Diaa Abd El-Moneim

Academic Editor

PLOS ONE

Reviewers' comments:

Reviewer's Responses to Questions

**Comments to the Author**

1. If the authors have adequately addressed your comments raised in a previous round of review and you feel that this manuscript is now acceptable for publication, you may indicate that here to bypass the “Comments to the Author” section, enter your conflict of interest statement in the “Confidential to Editor” section, and submit your "Accept" recommendation.

Reviewer #1: All comments have been addressed

Reviewer #2: All comments have been addressed

Reviewer #3: All comments have been addressed

2. Is the manuscript technically sound, and do the data support the conclusions?

Reviewer #1: Yes

Reviewer #2: Yes

Reviewer #3: Yes

3. Has the statistical analysis been performed appropriately and rigorously? 

Reviewer #1: Yes

Reviewer #2: Yes

Reviewer #3: Yes

4. Have the authors made all data underlying the findings in their manuscript fully available?

Reviewer #1: Yes

Reviewer #2: Yes

Reviewer #3: Yes

5. Is the manuscript presented in an intelligible fashion and written in standard English?

Reviewer #1: Yes

Reviewer #2: Yes

Reviewer #3: Yes

6. Review Comments to the Author

Reviewer #1: Dear Prof. Dr Editor-in-Chief of PLOS ONE,

I write you regarding Manuscript Number: Manuscript ID PONE-D-23-09518R1 entitled " Structure characteristics of mutation sites in two waxy alleles from Yunnan waxy maize (Zea mays L. var. certaina Kulesh) landraces" which submitted to PLOS ONE.

My decision is accepted for this work. The reason for that the authors answer all comments.

Thank you for suggesting me as a reviewer for this paper.

with best regards

Reviewer #2: Thanks to authors for improving their paper. The revised paper is better than the first copy, all comments were answered and corrected. I recommend publication for this paper

Reviewer #3: No comments, author has addressed all the questions.

The author has provided comprehensive and insightful explanations also for taking the time and effort to address all the questions with utmost clarity.

7. PLOS authors have the option to publish the peer review history of their article (what does this mean?). If published, this will include your full peer review and any attached files.

Reviewer #1: **Yes: **Khaled F M Salem

Reviewer #2: No

Reviewer #3: **Yes: **Dr. MUDASIR RASHID

---

## [Editor Report · Acceptance letter]

29 Aug 2023

PONE-D-23-09518R1 

Structure characteristics of mutation sites in two *waxy* alleles from Yunnan waxy maize (*Zea mays* L. var. *certaina* Kulesh) landraces 

Dear Dr. Wu:

I'm pleased to inform you that your manuscript has been deemed suitable for publication in PLOS ONE. Congratulations! Your manuscript is now with our production department. 

Kind regards, 

on behalf of

Dr. Diaa Abd El-Moneim 

Academic Editor

PLOS ONE